# Low-Resource Comparative Opinion Quintuple Extraction by Data Augmentation with Prompting

**Qingting Xu[1], Yu Hong[1][*], Fubang Zhao[2], Kaisong Song[3, 2], Yangyang Kang[2][*],**
**Jiaxiang Chen[1], Guodong Zhou[1]**

[1]School of Computer Science and Technology, Soochow University, Suzhou, China
[2]DAMO Academy, Alibaba Group, China [3] Northeastern University, China
{qtxu0801,tianxianer,xxfz56}@gmail.com
{fubang.zfb,kaisong.sks,yangyang.kangyy}@alibaba-inc.com
gdzhou@suda.edu.cn

## Abstract

Comparative Opinion Quintuple Extraction (COQE) aims to predict comparative opinion quintuples from comparative sentences. These quintuples include subject, object, shareable aspect, comparative opinion, and preference. The existing pipeline-based COQE method fails in error propagation. In addition, the complexity and insufficient amounts of annotated data hinder the performance of COQE models. In this paper, we introduce a novel approach called low-resource comparative opinion quintuple extraction by **D**ata **A**ugmentation with **P**rompting (**DAP**). Firstly, we present an end-to-end model architecture better suited to the data augmentation method from triplets to quintuples and can effectively avoid error propagation. Additionally, we introduce a data-centric augmentation approach that leverages the robust generative abilities of ChatGPT and integrates transfer learning techniques. Experimental results over three datasets (Camera, Car, Ele) demonstrate that our approach yields substantial improvements and achieves state-of-the-art results. The source code and data are publicly released at: https://github.com/qtxu-nlp/COQE-DAP.

## 1 Introduction

COQE is an essential subfield of Natural Language Processing (NLP). Its primary objective is to extract five specific components from comparative sentences, namely: subject, object, shareable aspect, comparative opinion, and preference, as defined in (Liu et al., 2021). For example, in the sentence "*Like the viewfinder, the Nikon D80 has the same sensor as the D200.*", "*Nikon D80*" and "*D200*" are respectively the subject and object entities, the aspect term is "*sensor*", the opinion word is "*same*", and the comparative preference is "*Equal*". COQE plays a crucial role in various applications, such as comparative opinion mining (Jindal and Liu, 2006b; Wang et al., 2010; Ma et al., 2020), sentiment analysis (Schouten and Frasincar, 2015; Zhang et al., 2022; Aftab et al., 2022), and customer satisfaction estimation (Ando et al., 2022).

Existing pipeline-based method (Liu et al., 2021) suffers from error propagation. The heavy reliance on extensive annotated data poses a bottleneck in the training process. To address the aforementioned issues, we propose a data augmentation method with prompting for low-resource COQE. Firstly, we propose a BERT-based (Devlin et al., 2018) end-to-end deep learning model as our backbone to avoid error propagation. Although existing LLMs such as ChatGPT possess rich linguistic knowledge and impressive generative capabilities, they encounter difficulties in generating satisfactory quintuple examples due to the inherent complexity of COQE. In this paper, we propose to develop a lightweight data augmentation, where the triple examples are required to be generated for augmentation, instead of the unabridged quintuple examples. It is relatively easy for ChatGPT to produce some qualified triple examples rather than quintuple. Additionally, we leverage these generated triple examples to warm up the end-to-end extraction model before training it over the benchmark quintuple dataset. To summarize, the main contributions of our work are as follows:

• We introduce an end-to-end model framework to suit data augmentation methods better and avoid error propagation. Additionally, we propose a two-stage data augmentation approach for low-resource COQE, leveraging the generative capabilities of ChatGPT and the transfer learning method.

• Experimental results demonstrate that our approach yields substantial improvements compared to the baseline and the current state-of-the-art model, resulting in a new highest performance on three COQE datasets. Furthermore, we conduct further analyses and supplementary experiments to verify the effectiveness of our approach.

---

[*] The authors contributed equally to this work and therefore are considered as co-corresponding authors.

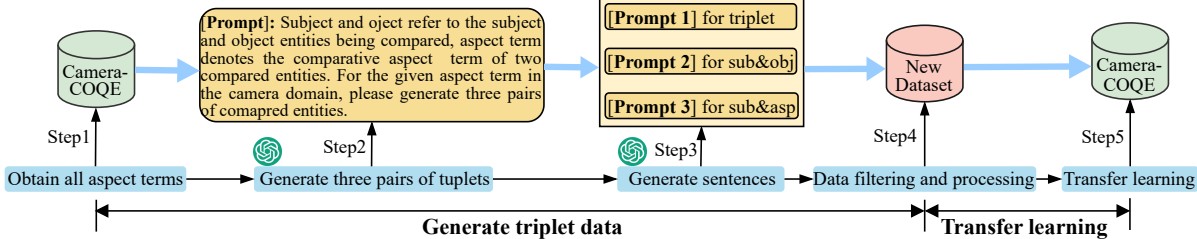

Figure 1: Overview of our approach. ChatGPT logo indicates steps requiring ChatGPT implementation.

## 2 Approach

### 2.1 Task Definition

COQE aims to predict quintuples $\{sub, obj, asp, op, pre\}$ from the given sentence $X = \{x_1, x_2, ..., x_n\}$ ($n$ denotes the number of tokens). *sub* and *obj* represent entities that serve as the subject and object in a comparison relation, respectively. *asp* refers to the shareable aspect term of two compared entities. Additionally, *op* denotes comments or opinions expressed regarding the aspect term of subject and object entities. There are four categories of preference (*pre*) to be considered, including *Better*, *Equal*, *Different* and *Worse*.

### 2.2 End-to-end Model

We utilize BERT (Devlin et al., 2018) as the sentence encoder. Using a BPE tokenizer (Sennrich et al., 2016), we obtain context-aware representations for each token in the input sentence $X$.

$$H = BERT(X) \qquad (1)$$

We employ a non-autoregressive (Guo et al., 2019) decoder to generate the quintuples, which remove the dependence on previous target tokens from the input of the decoder. Specifically, we randomly initialize a vector representation Q to represent a quintuple. In each decoder layer, we update the representation of Q using the formula (2). In this paper, we utilize a $l$-layer transformer decoder for non-autoregressive generation, and in our experiments, we set $l$ to 3.

$$Q_l = Decoder(H, Q_{l-1}) \qquad (2)$$

Given the output of the final decoder layer, we employ a classifier and four pointer networks to extract quintuples. Each pointer network is responsible for identifying the start and end position of one element within a quadruple. We calculate the classification probabilities using formula (3) and the extraction probabilities for the quadruple using formula (4).

$$p^c = softmax(W_c Q_l + b) \qquad (3)$$

$$p_i^e = softmax(V_i^\top tanh(W_e q_i^l + W_h H)) \qquad (4)$$

where $W_c$, $W_e$, $W_h$, $b$ and $V$ are all trainable parameters. $q_i^l$ is the $i$-th embedding output by the final decoder layer.

We optimize the combined objective function during training. $L_{total}$ comprises classification and extraction loss using the cross-entry loss function.

$$L_{total} = \sum_{n=1}^{N} \left( \log p^c + \sum_{k=1}^{K=8} \log p_k^e \right) \qquad (5)$$

where $N$ denotes the number of initialized Q, K denotes the number of loss functions computations required for a quadruple, which is equal to 8.

### 2.3 Data Augmentation for Transfer

Currently, there is a lack of datasets that are specifically annotated for subject, object and aspect in comparative sentences. In this paper, we introduce a data-centric method to leverage the rich linguistic knowledge within ChatGPT and further enhance COQE performance. ChatGPT generates a dataset containing triplets $\{sub, obj, asp\}$. In this section, we take the Camera dataset as an example. A multi-stage approach is needed to generate proper sentences for automatic annotation in the dataset. Building the triplet dataset involves four steps, as depicted in Figure 1.

• **Obtaining Aspect Terms** Firstly, we count the unique aspect terms separately from each dataset. The number of distinct aspect terms for three datasets is provided in Table 2.

• **Generating Triplets** We generate triplets based on the aspect terms obtained from step 1. Specifically, we invoke the ChatGPT API and design an appropriate prompt to generate three triplets.

| Kind | Prompt |
|---|---|
| $\{sub, obj, asp\}$ | *Please generate a new comparative sentence that compares or describes subject and object based on the given aspect term.* |

Table 1: The prompt for generating comparative sentences.

| Dataset | Camera | | Car | | Ele | |
|---|---|---|---|---|---|---|
| | #Sent | #Asp | #Sent | #Asp | #Sent | #Asp |
| Train | 2,114 | 510 | 2,269 | 500 | 2,304 | 421 |
| Dev | 529 | 160 | 568 | 163 | 576 | 141 |
| Test | 661 | 190 | 710 | 205 | 720 | 170 |

Table 2: Statistics in the training, dev and test sets. "Sent" and "Asp" indicate the total number of sentences and unique aspect terms for each dataset, respectively.

| Models | Camera | Car | Ele |
|---|---|---|---|
| $MS_{SVM+CRF}$ | 3.46 | 5.19 | 4.07 |
| $MS_{CRF}$ | 4.88 | 8.65 | 4.71 |
| $MS_{LSTM}$ | 9.05 | 10.28 | 14.90 |
| $MS_{BERT}$ | 13.36 | 29.75 | 30.73 |
| E2E | 19.21 | 35.02 | 36.66 |
| DAP | **21.06** | **36.13** | **39.24** |

Table 3: $F1$-scores for various COQE methods using exact strategy. The best scores are in bold.

• **Generating Sentences** Based on the statistics from the Camera dataset, 221 sentences contain triplets $\{sub, obj, asp\}$. In contrast, there are 202 sentences with only binary combinations such as $\{sub, obj\}$, 139 sentences with $\{sub, asp\}$, and 52 sentences with $\{obj, asp\}$. To ensure diversity in the generated sentences by ChatGPT, we prioritize the first three scenarios. We design specific prompts for each scenario. For example, in the first scenario, we establish the prompt as shown in Table 1.

• **Data Filtering and Processing** Despite the provided constraints, ChatGPT may still generate some samples that do not meet the specifications. Therefore, before automatic labeling, matching the generated sentences with the given triplets is essential. Only the sentences that successfully match the triplets should be labeled automatically.

We train the backbone model using newly constructed triplet data to obtain feature representations. Subsequently, we employ transfer learning techniques to fine-tune the gold quintuples based on the obtained representations.

## 3 Experiments

### 3.1 Datasets and Evaluation Metrics

**Datasets** We evaluate our method on three datasets. Camera is an English corpus (Liu et al., 2021). It builds upon the prior work of Kessler et al.(Kessler and Kuhn, 2014) by providing additional annotation for comparative sentences with comparative opinions and preferences. Besides, Liu et al. (Liu et al., 2021) construct two Chinese datasets specifically designed for comparative opinion quintuple extraction. They extend the COAE (Songbo Tan, 2013) dataset by building upon it and providing ad-

ditional annotations for data points regarding comparative opinions and preferences. The statistics of the three datasets are shown in Table 2.

**Evaluation Metrics** We evaluate all models through Precision ($P$), Recall ($R$), and $F1$ metrics. Additionally, we employ three matching strategies to evaluate prediction performance: exact-match (Ex), proportional-match (Pr) and binary-math (Bi) evaluation. The details of the three matching strategies are as follows:

$$Ex = \begin{cases} 0 & \exists\,(p_i \neq g_i) \\ 1 & \text{otherwise} \end{cases} \tag{6}$$

$$Pr = \begin{cases} 0 & \exists\,(g_i \cap p_i = \varnothing) \\ \frac{\sum_i len(g_i \cap p_i)}{\sum_i len(g_i)} & otherwise \end{cases} \tag{7}$$

$$Bi = \begin{cases} 0 & \exists\,(g_i \cap p_i = \varnothing) \\ 1 & otherwise \end{cases} \tag{8}$$

where, $g_i$ and $p_i$ denote the $i$-th element in the gold and predicted quintuple result, respectively. The index $i$ ranges from 1 to 5. We report the average performance based on three runs, utilizing shuffled random seeds for each run.

### 3.2 Compared Models

We compare with the following models:

$MS_{SVM+CRF}$ first propose comparative sentence identification. They utilize SVM (Cortes and Vapnik, 1995) for identifying comparative sentences and CRF (Lafferty et al., 2001) for extracting comparative elements (Jindal and Liu, 2006a).

| SOU → TAR | Metric | Model | P | R | $F1$ |
|---|---|---|---|---|---|
| Ele → Car | Ex | $MS_{BERT}$ | - | - | 23.44 |
| | | DAP | 33.19 | 27.10 | **29.83** (↑6.39) |
| | Pr | $MS_{BERT}$ | - | - | 30.49 |
| | | DAP | 47.84 | 39.06 | **42.99** (↑12.50) |
| | Bi | $MS_{BERT}$ | - | - | 31.64 |
| | | DAP | 50.50 | 41.23 | **45.39** (↑13.75) |
| Car → Ele | Ex | $MS_{BERT}$ | - | - | 24.42 |
| | | DAP | 34.31 | 28.91 | **31.38** (↑6.96) |
| | Pr | $MS_{BERT}$ | - | - | 31.61 |
| | | DAP | 52.70 | 44.42 | **48.21** (↑16.60) |
| | Bi | $MS_{BERT}$ | - | - | 32.85 |
| | | DAP | 55.97 | 47.18 | **51.20** (↑18.35) |

Table 4: Results of our proposed model in the cross-domain setting. The mark "-" indicates the results were not presented in (Liu et al., 2021)'s work.

**MS**$_{CRF}$ employ a CRF-based model for comparative sentence identification and comparative element extraction (Wang et al., 2015).

**MS**$_{LSTM}$ introduce a multi-stage framework utilizing LSTM as the text encoder. Firstly, they identify comparative sentences and extract comparative elements. Subsequently, they combine and filter these elements. Finally, they classify valid quadruples into four categories Liu et al. (2021).

**MS**$_{BERT}$ is a a modified version of MS$_{LSTM}$. For **MS**$_{BERT}$, Liu et al. (2021) choose BERT as the model's text encoder.

### 3.3 Main Results

We show the performance over three test sets in Table 3. It can be observed that DAP yields substantial improvements on three datasets compared to all the current SoTA or our backbone (E2E). In particular, even E2E is superior to the performance of MS$_{BERT}$. This also proves that the end-to-end model can effectively avoid error propagation. Besides, compared to the current SOTA results, DAP leads to $F1$ score improvements of 7.70%, 6.38%, and 8.51% on the Camera, Car, and Ele datasets, respectively. This shows that the performance of COQE can be effectively improved by introducing external knowledge contained in LLM.

### 3.4 Cross-Domain Experiments

We follow Liu et al. (2021) to evaluate the cross-domain generalization ability of our method. We conduct cross-domain experiments on two Chinese datasets, where cross-domain refers to using training and validation sets from the source domain (SOU) and a test set from the target domain (TAR). In Table 4, "Ele → Car" denotes electronic domain serves as the SOU, car domain is the TAR.

It can be observed that our method DAP yields

| Dataset | SST | PST |
|---|---|---|
| Camera | 8.5E-03 | 7.8 |
| Car | 3.4E-03 | 2.5 |
| Ele | 1.2E-02 | 4.8 |

Table 5: Results of two significance tests.

superior cross-domain experimental results surpassing previous COQE approaches, owing to our approach's generalization of data transfer.

### 3.5 Significance Test

We perform a statistical significance test (abbr., SST) to validate the reliability of our method. The sampling-based P-value (Johnson, 1999) is used as the metric for measuring significance levels. On the other hand, to provide a comprehensive insight into the significance, we conduct the practical significance test (abbr., PST) (Zhu et al., 2020), which is more reliable than SST. Cohen's D-value is used as the metric of PST. It is noteworthy that, in SST, the reported P-value below 0.05 (i.e., 5.0E-02) indicates significant improvement, otherwise insignificant (Dror et al., 2018). Similarly, in PST, the reported Cohen's D-value exceeds 1 indicates a significant improvement.

The results of the SST and PST are presented in Table 5. It can be found that the P-values of SST are lower than the threshold, while Cohen's D-value of PST is higher than the threshold. This demonstrates that DAP yields significant improvements.

## 4 Related Work

### 4.1 Comparative Sentence Analysis

Comparison-oriented information extraction has attracted considerable research interest. Jindal and Liu (2006a) first introduce the concept of comparative sentences and implement comparative sentence discrimination based on rules and SVM (Cortes and Vapnik, 1995). Park and Blake (2012) explore an extensive set of syntactic and semantic features, and employ three different classifiers to identify comparative sentences. The recent studies concentrate on fine-grained component analysis and parsing upon comparative sentences (Kessler and Kuhn, 2013; Arora et al., 2017; Ma et al., 2020). In particular, Liu et al. (2021) propose a novel task called comparative opinion quintuple extraction, which aims to extract quintuples from the given comparative sentences.

## 4.2 Data Augmentation

Data augmentation is a technique that expands and diversifies training datasets by applying various transformation or modification approaches to the existing data. Fadaee et al. (2017) use back-translation as a data augmentation method for generating synthetic parallel sentences, thereby enhancing the performance of low-resource neural machine translation systems. Wei and Zou (2019) use synonym replacement, random insertion, random deletion and synonym swapping to increase the diversity of training data. The approaches contribute to enhancing the robustness of text classifier. Chen and Qian (2020) develop a prototype generator for data augmentation, where internal and external prototypes are adopted.

## 5 Conclusion and Future Work

To avoid error propagation, we design an end-to-end model. Additionally, we propose a data-centric augmentation approach using the powerful generative capability of ChatGPT. The performance on three datasets achieves SoTA. Future work will focus on integrating existing annotated triplet data with automatically generated domain-specific data.

## Limitations

Despite achieving a new state-of-the-art performance, our model still has several limitations. One obvious limitation of the method is that the original three data sets contain a certain proportion of multiple comparison sentences, making predictions more difficult. In this paper, we only improve performance by introducing external knowledge and extracting the quintuple from the difficult to the easy. Future work can concentrate on tackling the COQE problem specifically from the perspective of multiple comparative sentences

## Acknowledgements

The research is supported by National Key R&D Program of China (2020YFB1313601), National Natural Science Foundation of China (62376182, 62076174, 62106039).

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
