# OpenReview forum: "Low-Resource Comparative Opinion Quintuple Extraction by Data Augmentation with Prompting"
_EMNLP/2023/Conference — EMNLP 2023 Findings_

### Official Review · Reviewer_1sq6 · 2023-07-24

**Soundness:** 4

**Excitement:**

4: Strong: This paper deepens the understanding of some phenomenon or lowers the barriers to an existing research direction.

**Paper Topic And Main Contributions:**

This paper introduces a novel Data Augmentation with Prompting approach to address two issues in comparative opinion quintuple extraction (COQE), namely applying Large Language Models (LLMs) directly to COQE has resulted in underwhelming performance and the complexity and insufficient amounts of annotated data hinder the performance of COQE models. Firstly, they propose an end-to-end model architecture better suited to the data augmentation method from triplets to quintuples. Additionally, they propose a two-stage data augmentation strategy combining the LLMs (ChatGPT) and transfer learning to further improve the performance of low-resource COQE. The experimental results validate the effectiveness and robustness.

**Questions For The Authors:**

Question A: The author did not provide a definition of what K represents in formula (5).

Question B: On page 3, line 202, I am not clear about the use of transfer learning, I think it is to conduct pre-training on the dataset generated by ChatGPT and then fine-tune on the COQE dataset. Please correct me if I have misunderstood.


**Reasons To Accept:**

In this paper, they propose a novel two-stage data augmentation method that leverage generative capabilities of ChatGPT and the transfer learning, which realizes the effect of data augmentation on COQE task. This has some inspiration on how to better combine other tasks with LLMs capabilities to improve the performance of target tasks.

**Reasons To Reject:**

no

**Reproducibility:**

4: Could mostly reproduce the results, but there may be some variation because of sample variance or minor variations in their interpretation of the protocol or method.

**Reviewer Confidence:**

4: Quite sure. I tried to check the important points carefully. It's unlikely, though conceivable, that I missed something that should affect my ratings.

---

> ### Author Rebuttal · Authors · 2023-08-28
>
> We are grateful for your insightful comments and questions.
>
>
>
> ----------Response to Questions ----------
>
>
> **Question #1:** The author did not provide a definition of what K represents in formula (5).
>
>
> **Response to #1:** Thank you for your suggestions. In our work, we employ a classifier and four pointer networks to extract quintuples. Each pointer network is responsible for identifying the start and end position of one element within a quadruple. For each pointer network, the value of K is set to 2. In formula (5), K denotes the number of loss functions computations required for a quadruple, which is equal to 8 (2*4) .
>
>
> **Question #2:** On page 3, line 202, I am not clear about the use of transfer learning, I think it is to conduct pre-training on the dataset generated by ChatGPT and then fine-tune on the COQE dataset. Please correct me if I have misunderstood.
>
> **Response to #2:** We appreciate your opinion, and it is absolutely correct. We first conduct pre-training on the dataset generated by ChatGPT, and then utilize transfer learning to fine-tune our model on the original COQE dataset.
>
>
> Thanks again for your constructive suggestions, and we will add more descriptions to solve any confusion in the extended version.

---

### Official Review · Reviewer_ttGM · 2023-08-02

**Soundness:** 3

**Excitement:**

3: Ambivalent: It has merits (e.g., it reports state-of-the-art results, the idea is nice), but there are key weaknesses (e.g., it describes incremental work), and it can significantly benefit from another round of revision. However, I won't object to accepting it if my co-reviewers champion it.

**Paper Topic And Main Contributions:**

This paper proposes an LLMs-based model for low-resource comparative opinion quintuple extraction. The author design a two-stage data augmentation strategy to improve the performance of COQE. Particularly, they use ChatGPT to generate triplet datasets and then use transfer learning. The experiment on three benckmark datasets shows their model outperforms the baseline.

**Questions For The Authors:**

1) How about the performance of ChatGPT? ChatGPT is not good at COQE, so why use it to generate data?
2) How about the performance of other data augmentation metheds and COQE methods?

**Reasons To Accept:**

1) The authors propose an LLMs-based data augmentation strategy for low-resource comparative opinion quintuple extraction.
2) This paper is well organized and easy to follow.

**Reasons To Reject:**

1) The existing baselines for Comparative Opinion Quintuple Extraction and data Augmentation are missing in the experiments.
2) The motivation of this paper is not clear. Why do they use ChatGPT to generate augmentation data? How about other data augmentation metheds?
3) Ablation studies are missing.

**Reproducibility:**

4: Could mostly reproduce the results, but there may be some variation because of sample variance or minor variations in their interpretation of the protocol or method.

**Reviewer Confidence:**

4: Quite sure. I tried to check the important points carefully. It's unlikely, though conceivable, that I missed something that should affect my ratings.

---

> ### Author Rebuttal · Authors · 2023-08-28
>
> We appreciate your insightful comments and promise to improve this paper in the final version.
>
>
>
> ----------Response to Comments----------
>
>
> **Comment #1:** The existing baselines for Comparative Opinion Quintuple Extraction and data augmentation are missing in the experiments.
>
>
> **Response to #1:** We realize your concern and regard it as the reason why there is only one SoTA approach involved for comparison. In fact, COQE is a new task which was proposed recently [1]. As a result, it hasn’t yet been sufficiently studied. To our best knowledge, the methodology that was published before this submission only comes from Liu et al. (2021)’s work [1]. Therefore, we concentrate on the comparison among our E2E (end-to-end) model, our enhanced version by data augmentation and Liu et al. (2021)’s Pipeline model [1]. We acknowledge that the COQE performance of some other neural models (e.g., LSTM and CRF) has been reported, though they are reported by Liu et al. (2021) themselves with the role of possible baselines. Due to the significantly worse performance of these conventional models compared to Liu et al. (2021)’s Pipeline model, we chose to omit them in our study. We feel sorry to simplify the comparison process, and provide the unabridged comparison results in Appendix behind these replies (Table 1). We also promise that we will expand the comparison experiments in the extended paper, after investigating the related work that may emerge recently.
>
>
> **Comment #2:** The motivation of this paper is not clear.
>
>
> **Response to #2:** We realize your concern and will respond to it from two aspects as below.
>
>
> * Error propagation: The pipeline model works in the way of progressively detecting and extracting the elements of each quintuple, step by step. This easily causes the error propagation problem. To address the issue, we propose to fulfill the quintuple extraction within an end-to-end framework, and develop the corresponding model.
>
>
> * Data sparsity: There is a limited annotated resource for COQE task, which leads to a problem of data sparsity. Data augmentation is potentially effective for addressing the issue. However, the existing large language models (LLM) like ChatGPT fail to generate satisfactory quintuple examples for augmentation due to the high complicacy of COQE. As a result, the conventional data augmentation approaches cannot be applied directly for COQE. To address the issues, we propose to develop a lightweight data augmentation, where the triple examples are required to be generated for augmentation, instead of the unabridged quintuple examples. It is relatively easy for LLM to produce some qualified triple examples rather than quintuple. Our approach uses the generated triple examples to warm up the end-to-end extraction model before training it over the benchmark quintuple dataset.
>
>
> **Comment #3:** Why do they use ChatGPT to generate augmentation data? How about other data augmentation methods?
>
>
> **Response to #3:** Data sparsity continues to be a significant concern, mainly due to the limited availability of annotated resources for the COQE task. ChatGPT has strong generative ability and easily produce high-quality data without the requirement of heavy manual efforts. The data generated by ChatGPT can introduce new knowledge and enhance the diversity of the dataset. Building on this, we utilize the data generated by ChatGPT in conjunction with transfer learning to enhance the performance of our model.
>
> Common data augmentation techniques, such as random insertion, deletion, and replacement, often introduce changes to the original semantics of sentences. Diverging from conventional data augmentation techniques, we adopt the approach of Karimi et al. [2] by randomly inserting a specified proportion of punctuation marks into sentences (referred by AEDA). This method maintains the original word order while altering the positions of words within the sentence, effectively enhancing the model's robustness. We show the performance of employing the AEDA method on the COQE task in Appendix behind these replies (Table 2). Compared to the popular AEDA method, our method achieves 1.1% improvements in average on Car and Ele datasets.
>
>
> **Comment #4:** Ablation studies are missing.
>
>
> **Response to #4:** Actually, we have conducted the ablation study. We propose a basic end-to-end model E2E and a data-augmentation based full model DAP. The results have been displayed in Table 2 (see lines 200-201 in our paper). These results are equivalent to the ablation experiments. We will highlight this part in our final version.
>
>
>
> ----------Response to Questions----------
>
>
> **Question #1:** How about the performance of ChatGPT?
>
>
> **Response to #1:** We conduct an analysis of directly employing ChatGPT for the COQE task, the results are illustrated in Figure 2 (lines 270-271). As you can see from Figure 2, even given well-designed prompt and instruction, implementing COQE tasks directly with ChatGPT has much lower performance than our DAP approach.
>
>
> **Question #2:** ChatGPT is not good at COQE, so why use it to generate data?
>
>
> **Response to #2:** Although ChatGPT has demonstrated success in other extraction tasks, its performance in COQE is notably inadequate. We attribute it to the fact that COQE consists of the product-domain data which are not fully used by ChatGPT. Additionally, COQE contains comparative opinions and proves to be more challenging than than aspect-based sentiment analysis. Generating quintuples directly poses a challenge, while obtaining triples is comparatively easier. This is the first work that discusses the methodology of utilizing ChatGPT in COQE, which paves the way for further research in NLP community.
>
> **Question #3:** How about the performance of other data augmentation methods and COQE methods?
>
>
> **Response to #3:** We conduct experiments comparing our approach with other data augmentation method AEDA (citation [2]), and the results are presented in Table 2. From the results, we can find our DAP outperforms AEDA because of COQE-specific data augmentation strategy.
>
>
>
> ----------Appendix----------
>
>
> **Table 1. F1-scores for various COQE methods.**
>
>
> | Models           | Camera | Car   | Ele   |
> | ---------------- | ------ | ----- | ----- |
> | Multi-Stage-CRF  | 3.46   | 5.19  | 4.07  |
> | Joint-CRF        | 4.88   | 8.65  | 4.71  |
> | Multi-Stage-LSTM | 9.05   | 10.28 | 14.90 |
> | Pipeline         | 13.36  | 29.75 | 30.73 |
> | DAP              | 14.96  | 36.13 | 39.24 |
>
>
> **Table 2. Comparison results with other data augmentation methods.**
>
>
> | Model    | Car   | Ele   |
> | -------- | ----- | ----- |
> | Pipeline | 29.75 | 30.73 |
> | AEDA     | 35.02 | 38.13 |
> | DAP      | 36.13 | 39.24 |
>
>
> Thanks again for all your constructive suggestions and we promise to solve all these problems in the final version.
>
>
>
> ----------Reference----------
>
>
> [1] Ziheng Liu, Rui Xia, and Jianfei Yu. 2021. Comparative opinion quintuple extraction from product reviews. In EMNLP.
>
> [2] Akbar Karimi, Leonardo Rossi, Andrea Prati. 2021. AEDA: An Easier Data Augmentation Technique for Text Classification. In EMNLP.

---

### Official Review · Reviewer_Azi6 · 2023-08-07

**Typos Grammar Style And Presentation Improvements:** 1. Line 206-209 We train the backbone…
**Soundness:** 2

**Excitement:**

3: Ambivalent: It has merits (e.g., it reports state-of-the-art results, the idea is nice), but there are key weaknesses (e.g., it describes incremental work), and it can significantly benefit from another round of revision. However, I won't object to accepting it if my co-reviewers champion it.

**Paper Topic And Main Contributions:**

This paper presents a data augmentation method using ChatGPT for comparitive opinion quintuple extraction. They also introduce an end-to-end model that benefits from such data augmentation. Although experiments show that generated data is clearly beneficial, the proposed model is not much convincing.

**Questions For The Authors:**

A. Line 128 - Where is Wt used?


**Reasons To Accept:**

- Easy to understand
- Positive results

**Reasons To Reject:**

This paper presents two things:
1. A model that does Comparitive Opinion Quintuple Extraction
2. Data augmentation pipeline using ChatGPT.

A. The significance of the data augmentation is shown via experiments but I am not convinced by the method.
B. Moreover, there is only one baseline that is considered. I would like to see more baselines being compared.
C. Can you also add the generated data to the "pipeline" and report its performance to see the improvement?


**Reproducibility:**

4: Could mostly reproduce the results, but there may be some variation because of sample variance or minor variations in their interpretation of the protocol or method.

**Reviewer Confidence:**

3: Pretty sure, but there's a chance I missed something. Although I have a good feel for this area in general, I did not carefully check the paper's details, e.g., the math, experimental design, or novelty.

---

> ### Author Rebuttal · Authors · 2023-08-28
>
> We appreciate your insightful comments.
>
>
>
> ----------Response to Comments----------
>
>
> **Comment #1:** The significance of the data augmentation is shown via experiments but I am not convinced by the method.
>
>
> **Response to #1:** We realize your concern and will respond to it from two aspects, including our motivation and the corresponding solution.
>
>
> * Motivation: There is a limited annotated resource for COQE task, which leads to a problem of data sparsity. Data augmentation is potentially effective for addressing the issue. However, the existing large language models (LLM) like ChatGPT fail to generate satisfactory quintuple examples for augmentation due to the high complicacy of COQE. As a result, the conventional data augmentation approaches cannot be applied directly for COQE.
>
>
> * Solution: To tackle the issues, we propose to develop a lightweight data augmentation, where the triple examples are required to be generated for augmentation, instead of the unabridged quintuple examples. It is relatively easy for LLM to produce some qualified triple examples rather than quintuple. Our approach uses the generated triple examples to warm up the end-to-end extraction model before training it over the benchmark quintuple dataset. In Section 2.3, we detail the instruction and prompt that we used for generating triple examples. In Table 2 (lines 200-201) of our paper, we compare the SoTA model (Pipeline) with our two models, including the end-to-end backbone (referred by E2E) and the enhanced version by augmentation (referred by DAP). It can be observed that substantial improvements are obtained by DAP, compared to E2E. This demonstrates the positive effectiveness of triple-example based data augmentation.
>
>
> In addition, we realize that your concern may attribute to the reliability of P-value based statistical significance test. Therefore, in order to provide a comprehensive insight into the significance, we conduct the practical significance test (PST) [2], which is more reliable than statistical significance test (SST) [1]. We list the test results of both PST and SST as below (namely Table 1 and Table 2), and hope they facilitate the verification of significance. It is noteworthy that, in SST, the reported P-value corresponds to a significant improvement only if it is smaller than 0.05, while in PST, the reported Cohen’s D-value corresponds to a significant improvement only if it is a much higher value than 1.
>
>
> **Comment #2:** Moreover, there is only one baseline that is considered. I would like to see more baselines being compared.
>
>
> **Response to #2:** We recognize your perspective and consider it to be the justification for including just one state-of-the-art approach for comparison. In fact, COQE is a new task which was proposed recently [1]. As a result, it hasn’t yet been sufficiently studied. To our best knowledge, the only methodology published prior to this submission originates from the work of Liu et al. (2021) [1]. Therefore, we concentrate on the comparison among our E2E (end-to-end) model, our enhanced version by data augmentation and Liu et al. (2021)’s Pipeline model [1]. We acknowledge that the COQE performance of some other neural models (e.g., LSTM and CRF) has been reported, though they are reported by Liu et al. (2021) themselves with the role of possible baselines. Due to the significantly worse performance of these conventional models compared to Liu et al. (2021)’s Pipeline model, we chose to omit them in our study. We feel sorry to simplify the comparison process, and provide the unabridged comparison results in Appendix behind these replies (Table 3). We also promise that we will expand the comparison experiments in the extended paper, after investigating the related work that may emerge recently.
>
>
> **Comment #3:** Can you also add the generated data to the "pipeline" and report its performance to see the improvement?
>
>
> **Response to #3:** We understand your concern and assure you that we did consider the possibility of adding such data upon the Pipeline model. However, our data augmentation strategy cannot be directly applied upon it. It is because of the following reasons.
>
> * Reason 1: First, we generate triple examples instead of quintuples, as explained in the response to Comment #1. Though, such triple examples aren’t friendly to the Pipeline model. In other words, the input of the Pipeline model is fixed, which cannot be dynamically changed.
>
> * Reason 2: Second, we did work on the enhancement of the pipeline model by adding the generated quintuples. However, the model performance decreases by 2.5% in average due to a large amount of noises and redundant information in the generated quintuples. We will add more discussions about this in the extended version of our paper and explore more extensible data augmentation strategy in the future.
>
>
>
> ----------Response to Questions ----------
>
>
> **Question #1:** Line 128 - Where is Wt used?
>
>
> **Response to #1:** In line 128, "Wt" should be corrected as "Wc". We will correct this typo.
>
>
>
> ----------Response to Typos Grammar Style And Presentation Improvements ----------
>
>
> **Suggestion #1:** Line 206-209 We train the backbone model using newly constructed data.
>
>
> **Response to #1:** Thanks you for reminding us about the presentation. What we intend to present in the sentence is about “We train the backbone model using the newly constructed triple data”. We will modify this description, so as to make is easy to follow.
>
>
>
> ----------Appendix----------
>
>
> **Table 1. The results of SST (P-value is considered).**
>
>
> |            | Camera | Car    | Ele   |
> | ---------- | ------ | ------ | ----- |
> | E2E VS DAP | 0.0085 | 0.0034 | 0.012 |
>
>
> **Table 2. The results of PST (Cohen's D-value is considered).**
>
>
> |            | Camera | Car  | Ele  |
> | ---------- | ------ | ---- | ---- |
> | E2E VS DAP | 4.32   | 1.31 | 2.62 |
>
>
> **Table 3. F1-scores for various COQE methods.**
>
>
> | Models           | Camera | Car   | Ele   |
> | ---------------- | ------ | ----- | ----- |
> | Multi-Stage-CRF  | 3.46   | 5.19  | 4.07  |
> | Joint-CRF        | 4.88   | 8.65  | 4.71  |
> | Multi-Stage-LSTM | 9.05   | 10.28 | 14.90 |
> | Pipeline         | 13.36  | 29.75 | 30.73 |
> | DAP              | 14.96  | 36.13 | 39.24 |
>
>
>
> ----------Reference----------
>
>
> [1] Douglas H Johnson. The insignificance of statistical significance testing. In the journal of wildlife management 1999.
>
> [2] Haotian Zhu, Denise Mak, Jesse Gioannini, Fei Xia. NLPStatTest: A Toolkit for Comparing NLP System Performance. In ACL 2020.
>
> [3] Rotem Dror, Gili Baumer, Segev Shlomov, and RoiReichart. The hitchhiker’s guide to testing statistical significance in natural language processing. In ACL 2018.
>
> [4] Ziheng Liu, Rui Xia, and Jianfei Yu. Comparative opinion quintuple extraction from product reviews. In EMNLP 2021.

---

### Official Review · Reviewer_wHff · 2023-08-11

**Soundness:** 3

**Excitement:**

3: Ambivalent: It has merits (e.g., it reports state-of-the-art results, the idea is nice), but there are key weaknesses (e.g., it describes incremental work), and it can significantly benefit from another round of revision. However, I won't object to accepting it if my co-reviewers champion it.

**Paper Topic And Main Contributions:**

This paper is about Comparative Opinion Quintuple Extraction (COQE) where a two-stage data augmentation strategy is proposed. Furthermore, experiments were conducted to verify the effectiveness of the method on 3 available COQE datasets

**Reasons To Accept:**

The paper provides a novel concept to predict the triplets - subject, object and aspect. It depicts the comparisons between ChatGPT and fine-tuned BERT model while obtaining the triplets



**Reasons To Reject:**

The Data Augmentation with prompting (DAP) technique is not clearly explained how it compares with the benchmark and outperforms subsequently.

The organization of the paper sections could have been a little better.

**Reproducibility:**

3: Could reproduce the results with some difficulty. The settings of parameters are underspecified or subjectively determined; the training/evaluation data are not widely available.

**Reviewer Confidence:**

4: Quite sure. I tried to check the important points carefully. It's unlikely, though conceivable, that I missed something that should affect my ratings.

---

> ### Author Rebuttal · Authors · 2023-08-28
>
> We appreciate your insightful comments and promise to improve this paper in the revised version.
>
>
>
> ----------Response to Comments----------
>
>
> **Comment #1:** The Data Augmentation with prompting (DAP) technique is not clearly explained how it compares with the benchmark and outperforms subsequently.
>
>
> **Response to #1:** I sincerely sorry for any inconvenience caused by our failure to adequately clarify the Data Augmentation with prompting (DAP) technique in the paper. In fact, we introduce  two main innovations in this paper: 1) Compared with existing pipeline methods which fail in the issue of error propagation, we propose an end-to-end extraction framework (i.e., **E2E**). 2) Faced with the tricky low-resource problem in COQE, we introduce a data-centric augmentation approach that leverages the robust generative abilities of ChatGPT and integrates transfer learning techniques (i.e., **DAP**). COQE is a new task which was proposed recently [1]. As a result, it hasn’t yet been sufficiently studied. To our best knowledge, the only methodology published prior to this submission comes from Liu et al. (2021)’s work [1]. Therefore, we concentrate on the comparison among our E2E model, our enhanced version DAP and Liu et al. (2021)’s Pipeline model [1]. We show the experimental result in Appendix behind these replies (Table 1). It can be concluded that the data augmentation method (DAP) proposed in this paper not only demonstrates improvements over the baseline model (E2E) but also exhibits significant advancements when compared to the current state-of-the-art (**Pipeline**) methods.
>
>
> **Comment #2:** The organization of the paper sections could have been a little better.
>
>
> **Response to #2:** We acknowledge that the organization of the paper could be improved for better clarity and coherence. Based on your suggestions, we intend to restructure our paper sections. For instance, we propose the incorporation of a new Section 2.2, dedicated to offering a comprehensive overview of the overall architecture of DAP. We appreciate your suggestions and  promise to make the necessary revisions in the final version.
>
> ----------Appendix----------
>
>
>
> **Table 1. F1-scores for various COQE methods.**
>
>
> | Models           | Camera | Car   | Ele   |
> | ---------------- | ------ | ----- | ----- |
> | Pipeline         | 13.36  | 29.75 | 30.73 |
> | E2E  | 12.56  |35.02 | 36.66 |
> | DAP              | 14.96  | 36.13 | 39.24 |
>
>
> We sincerely appreciate your thoughtful insights and all the constructive suggestions. We will strictly follow your suggestions to improve the paper.
>
>
>
> ----------Reference----------
>
>
> [1] Ziheng Liu, Rui Xia, and Jianfei Yu. Comparative opinion quintuple extraction from product reviews. In EMNLP 2021.

---

### Meta-Review · Area_Chair_WJ6F · 2023-09-24

**Recommendation:** 4

**Metareview:**

Authors present the low-resource comparative opinion quintuple extraction by Data Augmentation with Prompting (DAP). Authors use a two-stage data augmentation strategy which first guides ChatGPT to generate triplet datasets and then employs transfer learning.

One of the 4 reviewers did not acknowledge the reading of the rebuttal.
For the soundness, reviewers give 2 (borderline), 3(good), 3(good), and 4(strong). For the excitement, 3 reviewers give 3 (ambivalent) and 1 gives 4 (strong). Some discussions authors did with the reviewer giving 2 were post-rebuttal. The response from authors has a reasonable explanation for the post-rebuttal question, as well as to the other questions. I have taken all these issues into account.

---

### Decision · Program_Chairs · 2023-10-07

**Decision:**

Accept-Findings

**Comment:**

Authors present the low-resource comparative opinion quintuple extraction by Data Augmentation with Prompting (DAP). Authors use a two-stage data augmentation strategy which first guides ChatGPT to generate triplet datasets and then employs transfer learning.

One of the 4 reviewers did not acknowledge the reading of the rebuttal.
For the soundness, reviewers give 2 (borderline), 3(good), 3(good), and 4(strong). For the excitement, 3 reviewers give 3 (ambivalent) and 1 gives 4 (strong). Some discussions authors did with the reviewer giving 2 were post-rebuttal. The response from authors has a reasonable explanation for the post-rebuttal question, as well as to the other questions. I have taken all these issues into account.